# Resilience in Lower Grade Glioma Patients

**DOI:** 10.3390/cancers14215410

**Published:** 2022-11-02

**Authors:** Ellen Fröhlich, Claudia Sassenrath, Minou Nadji-Ohl, Meike Unteroberdörster, Stefan Rückriegel, Christian von der Brelie, Constantin Roder, Marie-Therese Forster, Stephan Schommer, Mario Löhr, Andrej Pala, Simone Goebel, Dorothee Mielke, Rüdiger Gerlach, Mirjam Renovanz, Christian Rainer Wirtz, Julia Onken, Marcus Czabanka, Marcos Soares Tatagiba, Veit Rohde, Ralf-Ingo Ernestus, Peter Vajkoczy, Oliver Gansland, Jan Coburger

**Affiliations:** 1Department of Neurosurgery, University of Ulm, 89312 Günzburg, Germany; 2Department of Social Psychology, Institute of Psychology and Education, Faculty of Engering, Informatics and Psychology, University of Ulm, 89312 Günzburg, Germany; 3Department of Neurosurgery, Klinikum Stuttgart, 70174 Stuttgart, Germany; 4Department of Neurosurgery, Charité—Universitätsmedizin Berlin, 12200 Berlin, Germany; 5Department of Neurosurgery, University of Würzburg, 97080 Würzburg, Germany; 6Department of Neurosurgery, Johanniter—Kliniken Bonn, 53177 Bonn, Germany; 7Department of Neurosurgery, University of Tübingen, 72076 Tübingen, Germany; 8Department of Neurosurgery, University of Frankfurt, 60528 Frankfurt am Main, Germany; 9Department of Psychology, University of Kiel, 24118 Kiel, Germany; 10Department of Neurosurgery, University of Göttingen, 37075 Göttingen, Germany; 11Department of Neurosurgery, Helioskliniken Erfurt, 99089 Erfurt, Germany; 12Department of Neurology and Interdisciplinary Neuro-Oncology, Hertie Institute for Clinical Brain Research, Eberhard-Karls University of Tübingen, Otfried-Müller-Straße 27, 72076 Tübingen, Germany; 13Center for Neuro-Oncology, Comprehensive Cancer Center Tübingen-Stuttgart, University Hospital of Tuebingen, Eberhard Karls University of Tübingen, 72076 Tübingen, Germany; 14Department of Neurosurgery, University Hospital Tübingen, Eberhard Karls University Tübingen, Hoppe-Seyler-Str. 3, 72076 Tübingen, Germany

**Keywords:** resilience, lower grade glioma, diffuse astrocytoma, oligodendroglioma, RS-13, distress, internalized stigmatization, ISBI, occupation, pain

## Abstract

**Simple Summary:**

Current data show that resilience is an important factor in cancer patients’ well-being. We explored the resilience of patients with lower grade glioma (LGG) and the potentially influencing factors. Our data indicate that stigmatization and the functional status are significantly associated with the patients’ resilience. These factors should be identified and targeted therapeutically in clinical routines.

**Abstract:**

Current data show that resilience is an important factor in cancer patients’ well-being. We aim to explore the resilience of patients with lower grade glioma (LGG) and the potentially influencing factors. We performed a cross-sectional assessment of adult patients with LGG who were enrolled in the LoG-Glio registry. By phone interview, we administered the following measures: Resilience Scale (RS-13), distress thermometer, Montreal Cognitive Assessment Test for visually impaired patients (MoCA-Blind), internalized stigmatization by brain tumor (ISBI), Eastern Cooperative Oncological Group performance status (ECOG), patients’ perspective questionnaire (PPQ) and typical clinical parameters. We calculated correlations and multivariate regression models. Of 74 patients who were assessed, 38% of those showed a low level of resilience. Our results revealed significant correlations of resilience with distress (*p* < 0.001, −0.49), MOCA (*p* = 0.003, 0.342), ECOG (*p* < 0.001, −0.602), stigmatization (*p* < 0.001, −0.558), pain (*p* < 0.001, −0.524), and occupation (*p* = 0.007, 0.329). In multivariate analyses, resilience was negatively associated with elevated ECOG (*p* = 0.020, β = −0.383) and stigmatization levels (*p* = 0.008, β = −0.350). Occupation showed a tendency towards a significant association with resilience (*p* = 0.088, β = −0.254). Overall, low resilience affected more than one third of our cohort. Low functional status is a specific risk factor for low resilience. The relevant influence of stigmatization on resilience is a novel finding for patients suffering from a glioma and should be routinely identified and targeted in clinical routine.

## 1. Introduction

Gliomas are the most frequently treated primary intracranial neoplasms [1,2]. Lower grade gliomas (LGG) WHO grade II and III are slowly growing gliomas which inevitably result in recurrence and malignant progression. Based on the low incidence of 0.7 per 100,000 persons [3] and the relatively long overall survival rates, the general evidence and treatment recommendations as well as knowledge regarding the sequelae for quality of life is scarce [4].

Patients with identical tumors and treatment strategies may experience considerable differences regarding how severe they perceive the burden resulting from their disease [5]. In addition to medical treatment, the individuals’ ability to resist and adapt positively may play an important role in recovering from the severe disease. [6]. In this regard, resilience, which is defined as dynamic multi-modal construct referring to the individuals’ adaptive response to adversity likely plays an important role in recovering from stressful situations such as cancer [7,8,9]. Recent findings supporting the notion of resilience being meaningful in the recovery from severe diseases indicate that the stronger patients’ resilience is, the more emotional stability they display and the less discomfort they report. [10]. Moreover, a substantial interaction between the patients’ psycho-oncological distress and resilience has been found, and a corresponding model was proposed by Kumpfer (1999), proposing clinical stressors and patient-specific and generic elements as mediating factors [9].

Given the scarcity of the empirical evidence regarding the resilience of patients with LGGs, the present study aimed at assessing the clinically relevant influential factors on the resilience of patients with LGGs.

Apart from clinical factors which are referred to as stressors by Kumpfer et al. [9], we investigated the impact of additional, influencing patient-specific psychosocial factors such as functional status, internalized stigmatization by brain tumor as well as social support.

## 2. Materials and Methods

### 2.1. Patient Selection

The patients participating in our current study were selected from the LoG Glio-Registry’s pool of patients who were enrolled in the study between 2015 and 2021. The LoG Glio-Register (NCT02686229) encloses adult patients with radiologically suspected diagnosis of a lower grade glioma. A detailed study protocol was published previously [4].

In the current study, we included patients with diffuse gliomas of WHO grade II or III, with a complete 1st follow-up datasets from centers with 3 or more included patients. See Figure 1 for patient selection.

The histopathological diagnosis of patients in this study was performed according to the WHO 2016 classification. All of the patients in this study either had a biopsy or open surgery that confirmed the diagnosis.

The study patients were contacted and selected for the study if they additionally consented to a detailed phone interview. Only German-speaking patients were included. In the case of patients suffering from aphasia, a Montreal Cognitive Assessment Test (MoCA)Blind score of ≥11 and the possibility of participating in individual communication were mandatory for participation in the study.

### 2.2. Study Design

We applied a cross-sectional study design. A telephone interview was used to assess their cognitive ability, distress, resilience, performance, need for support and care from a patient’s perspective, stigma and pain. In case participants needed a break, the interview was split into two telephone calls. The patients’ demographic characteristics such as age, year of initial diagnosis and treatment were obtained from the LoG-Glio registry. Their educational level was dichotomized into higher education (school diploma qualifying for tertiary education) and lower education.

### 2.3. Questionnaires

MoCA-Blind:

We used the MoCA-Blind to evaluate the participants’ capacity to answer the study questions. This questionnaire measures memory, attention, concentration, language, abstraction, delayed recall, and orientation. For the phone interview, we used a pen to tap on a hard surface for every instance of a letter ‘A’ during the attention testing procedure.

The MoCA questionnaire was originally developed by Nasreddine in 2005 [11]. It allows for an initial evaluation of their cognitive ability. For the visually impaired patients, the test remained identical, but the visual, spatial and the naming tasks were removed [12]. A score below 19 points as considered as pathological.

NCCN Distress Thermometer:

The NCCN Distress Thermometer (DT) is a well-validated screening instrument to assess the psychological distress of intracranial tumor patients, which was developed by the National Comprehensive Cancer Network (NCCN) [13,14]. We used the cut-off score of ≥6 because of the higher specificity is used for neurosurgical cancer patients [14].

RS-13—Resilience Scale

The Resilience Scale 13 is a 3-factor measure covering overall resilience, acceptance and competence [10]. It is especially suitable for the medical field because it not only does it include factors such as personal competence to optimize resilience, but it also assesses the patients’ ability to accept and adapt to unchangeable circumstances.

ISBI-10—a brief internalized stigma of brain tumor inventory

We adapted the ISBI questionnaire which is based on the brief internalized stigma of mental illness inventory ISMI-10 [15], which was originally developed for people with mental disturbances. Accordingly, the German version was rephrased from a focus on schizophrenia to that of brain tumors, and one item was removed from the original questionnaire due to consistency reasons. Similar adaptations have already been performed in the context of the stigmatization of patients with rheumatoid arthritis [16]. The ISMI-10 represents a screening tool with a high-psychometric quality and good acceptance by the patients [15,17].

ECOG Performance Scale

The Eastern Cooperative Oncology Group (ECOG) Performance Scale has been a widely used and established test for more than 30 years. It is an assessment of ‘the actual level of function and capability for oncological patients to practice self-care’, further referred to as their functional status. It is an established prognostic factor for survival and quality of life [18].

PPQ—Patients’ Perspective Questionnaire

This questionnaire assesses the need for support and care from the patients’ perspective [19]. It is a typical patient-reported outcome measure (PROM), particularly addressing the patients’ perceptions of requested and received support.

In our model calculations, we included item 8 from the list of perceived support (‘Currently I feel sufficiently supported by others’, e.g., family, friends, physicians, etc.), measuring the general support on a Likert scale from 1 to 5 most adequately and consistently.

Numerical Rating Scale (NRS) for pain

A scale from 0 ‘no pain’ to 10 ‘worst imaginable pain’ was used to determine the existing pain and the intensity of it [20].

In addition, the typical clinical parameters regarding adjuvant treatment, occupation and relationship status were documented.

### 2.4. Statistical Analysis

Data analysis was performed using SPSS Version 28.0.1.0 (IBM Corp. Released 2022. IBM SPSS Statistics for Windows, Version 21.0. Armonk, NY, USA).

We applied an explorative data assessment. A Pearson correlation analysis was used to identify the correlations between the internal stigma (ISBI-10), resilience (RS-13), distress (DT), performance (ECOG), perceived sufficiency of received support by others (PPQ) and cognitive ability (MoCA-Blind) as well as pain (NRS) factors. The binary correlation with occupation was tested using Spearman’s ρ. No corrections were made for multiple testing.

Additionally, we tested the ISBI for internal consistency, and calculated the Cronbach’s alpha since the ISBI-10 score was first used for the glioma patients.

In order to assess the influencing factors on resilience, we regressed the RS-13 score using multinomial linear regression analyses on the existing pain (NRS), chemotherapy and radiation treatment, ECOG, MoCA-Blind, ISMI-10, gender, time since primary diagnosis of disease, age, perceived sufficiency of support by others, education and relationship status factors. Another multinomial linear regression analysis including the same predictors, but distress as a criterion was calculated. The results are displayed in the (Appendix A).

The current study was conducted according to the international Declaration of Helsinki. Approval from the local ethic committee was obtained.

## 3. Results

### 3.1. Sample

We assessed 74 patients with a median age of 43 years (min 21, max 67). Exactly 43.2% of the participants were female, and 56.8% of them were male. Exactly 55.4% of the patients had a high educational level, and 79.7% of them were in a relationship. Most of the participants were working (62.2%), but also 8.1% of them were on sick leave and 25.7% were retired. Table 1 shows the demographic characteristic of the included patients.

The median time since diagnosis was 4 years (min < 1, max 13). Table 2 shows characteristics of adjuvant treatment

### 3.2. Questionnaires

The descriptive results (overall frequencies and percentages) of the focal questionnaires (MoCa-Blind-Test; pathological Distress; RS-13; ECOG; ISMI-10, and Pain) are displayed in Table 3.

Notably, the MoCA-Blind test had a median score of 19 points (min.10; max 22), which indicates that our participants were cognitively able to understand and answer our questionnaires. The median of the distress level was four (min. 0, max 10). Twenty-nine patients (39.2%) had a pathologic distress level of six or above, from whom 55.2% had at least once contacted a psychiatrist, which often took place after or before surgery in the hospital.

The median result for social support (values obtained from the PPQ, item 8; not displayed in Table 3) was 4.3 (min. 2, max. 5), using a Likert scale from one to five.

Additional information is as follows: The resilience factor which was assessed via RS-13 had a median result of 67 points (min. 27; max 90). Out of 28 patients with a low (13–66 points), 48.3% of them had a pathologic DT score, from 16 with a moderate level of (67–72 points) resilience, 24.1% of them had a pathologic DT score from, and from 30 patients with a high level of resilience (73–91 points), 27.6% of them had a pathologic DT score. Additionally, 50% of the participants with a low level of resilience had a subnormal MoCA-Blind score.

Compared to the validated norm sample of the RS-13 by Leppert et al. (2008), 38% of our patients had a low resilience (<67 vs. 30% of norm sample patients).

The ECOG showed that 57% of the participants in our study had no performance issues. Twelve percent of them were able to carry out work of a light or sedentary nature. Nineteen percent of them could take care of themselves but could not work anymore. Eleven percent of them were confined to a bed or chair for over fifty percent of the waking hours and could not take care of themselves anymore. One percent of them were completely disabled.

In our study, the median score for the internal stigma factor, ISBI-10, was 16 points (min 11; max 28). It turned out that 88% of the participants had no to minimal internal stigmatization (result: 1.00–2.00), 8% of them suffered from a slight amount of it (result: 2.01–2.50), and 4% of them suffered from a moderate amount of (result: 2.51–3.00) internal stigma.

Of our 74 patients, 55 (74%) reported no pain. For the 19 patients with pain, the median level of the numeric rating scale (NRS) of pain was 4.8 (min 1; max 8).

In our study, 46 (62%) participants were in occupation. As Table 4 indicates, the results for patients with occupations regarding the MoCA-Blind, ECOG and NRS values were better when they were compared to the patients without occupations (Table 4).

### 3.3. Correlations

Table 5 and Table 6 display the correlations (both Pearson and Spearman-Rho, given that some variables represent continuous variables, and some represent categorical variables) of all of the focal variables. Specifically, resilience is negatively correlated with distress (*p* < 0.01, −0.49), ECOG (*p* < 0.001, −0.602), stigmatization (*p* < 0.001, −0.60), and pain (*p* < 0.001, −0.52). On the other hand, MoCA-Blind (*p* = 0.003, 0.34), occupation status (*p* = 0.005, 0.329) and social support (*p* < 0.020, 0.271) are positively correlated with resilience (Table 5). (See Appendix A for correlations of distress and the focal variables.)

### 3.4. Multivariate Model

Applying a multivariate approach, the results of the multinomial linear regression analyses are displayed in Table 7. Here, a low functional status and internalized stigma are negatively associated with resilience, and occupation status is marginally negatively associated with resilience. No associations were found for education, gender, age and time since diagnosis (Table 7). (See Appendix A for association of distress and the focal variables.)

## 4. Discussion

LGG is a slow-progressing disease, resulting inevitably, in malignancy and a limited life expectancy [21]. The patients’ resilience and positive adaptation to their condition might play a relevant role in the course of LGG, given that the ability to positively adapt is needed in addition to medical treatments when one is suffering from a palliative disease [6]. Moreover, a wide range of responses can be observed in the patients with identical diseases and treatments [5], whereas some patients put up with the disease and the psychological consequences, and they are able to work and have fulfilled lives, while others are affected by depression, are in pain, and their quality of life is diminished. Strengthening resilience could be a key to solving this apparent inconsistency as it affects the ability to adapt to stressful events [7,8]. It is important to consider that resilience may be influenced by clinical stressors and specific and generic elements that either lead to a gain or to a loss of psychological resources [9]. Increasing resilience should, thus, be of high priority in patient care. Accordingly, this research took a first step in exploratively assessing the potentially influencing factors on resilience within our patients’ cohort.

More than one third of patients showed a low resilience in our study, which is slightly lower than in the norm sample of Leppert et al. [11]. According to our results, education, gender and time since primary diagnosis did not have a significant impact on resilience. However, distress, pain, and low functional status were negatively associated with resilience, and occupation was positively associated with resilience. Interestingly, internal stigma was correlated with a lower resilience as well. In the multivariate analyses, only the functional status and stigmatization remained significantly associated with resilience. Interestingly, occupation was marginally negatively associated with resilience.

Even though only around 10% of the patients suffered from relevant internal stigmatization, it has a significant negative effect on resilience. The patients who were included in the study were often unaware that they suffered from self-stigma. In the literature, it has been shown for several diseases that internal stigma plays an important role and may subsequently negatively affect their health outcomes [22,23]. Hence, it is most likely that it also plays a vital role in brain tumors, especially since brain tumors may affect the patients’ memory, language, personality and behavior like no other oncological disease does. Notably, at time of writing this article, to our knowledge no publication exists that addresses this question in brain tumors.

In our study, we discovered that every fourth patient was suffering from pain which has a negative impact on their resilience, and subsequently, it also had a negative effect on their quality of life and emotional stability [10]. Especially regarding the group with LGG, physicians are not aware of this problem, whereas according to a review that was published by Taillibert et al. [24] 50% of glioma patients suffer from pain, and the current data that deal with this specific problem are rare—only one of them discussed pain specifically. Here, the authors reported that even more than 50% of glioma patients suffer from pain [25]. Interestingly, a Pearson correlation identified a significant negative association between pain and resilience. In our multivariate analyses, however, we did not find a significant association. One potential explanation here could be the highly significant association with distress as shown in the Appendix A.

Occupation was correlated positively with resilience. Yet, when controlling for other influencing variables in the multivariate model, we found a marginally significant negative association with resilience. Occupation is important for psychological health since it provides a daily rhythm, social status, income and also maintains the ability to solve problems [26], and while it is significantly associated with reduced distress, it is negatively associated with resilience. This underlines the distinct differences of both of the concepts. Occupation may be an additional burden for some patients that may decrease the patients’ resources to build up or strengthen their resilience. Thus, returning to work should be encouraged only as long as it appears to be a realistic option and does not contribute to the patients’ burden and efforts to adapt to their current condition.

Furthermore, our data show that a low functional status according to the ECOG scores is negatively associated with resilience. According to the model by Kumpfer et al., neurological deficits or fatigue are relevant stressors negatively influencing resilience. Additionally, a recent publication by Yang et al. showed a strong association between performance status and resilience [27].

In LGG, the patient’s psycho-oncological distress is a relevant [28], and often, it is an underestimated burden. It includes difficult emotional experiences of the psychological, social or spiritual kind [29]. It is often associated with depression, affecting over 20% of treated cancer patients [30], and has a negative influence on long term survival [31]. It correlates negatively with resilience and is defined as pathological stress when a DT score of six or higher is reached [14]. Distress does not have to be emotional; it can also be financial, family-related, physical or a mixture of all of them. It is essential to comprehend distress according to its cause. Both of the terms, resilience and distress, are closely related because they affect the coping responses and personal reactions of patients regarding their cancer disease [7]. In our data, we observed both of these in direct correlations as well as in the multivariate models that stated that resilience and distress share common influencing parameters. However, especially in the multivariate model, they reveal distinctly different features. While distress is significantly associated with pain, relationship status, occupation and only slightly with functional status, resilience is only affected by functional status and stigmatization.

### 4.1. Clinical Implications and Future Perspectives

The specific features of resilience and distress should be considered during scheduled follow-up visits at neurooncological outpatient departments. Particularly, stigmatization is highly underdiagnosed. A screening and consecutive anti-stigma intervention, as it already exists for mental disturbances, should be developed for brain tumors as well [32].

Especially, patients with a low or decreasing functional status seem to be at risk of having a weak level of resilience. These deficits are relatively well recognizable when they are compared to the above-mentioned stigmatization factor. We should be more aware of the options to increase resilience such as self-empowerment programs to improve coping mechanisms or patient associations. Furthermore, the early involvement of social workers and/or psychologists might result in stronger resilience and finally even in better survival rates.

After their discharge from the hospital, the patients’ living situation often changes due to issues such as possible physical disabilities, an uncertain future, or probably financial difficulties, for example. Accordingly, 70% of our study participants (52 of 74) indicated that an institutionalized professional support targeted at cancer patients would be of help. In the case of palliative patients, such a support has already been implemented with great success [33]. Specialized outpatient palliative care (SOPC) has become an important component of palliative care concepts in Germany. However, patients are often referred to them too late. Early contact to these centers might also improve the patients’ distress and increase their resilience, potentially enabling them a living at home until their death.

It could be shown that resilience has a positive influence on the health outcome in breast cancer patients concerning their health-related quality of life and a reduced risk of depression [34,35]. Further research is needed to assess whether there is an indirect or even a direct influence on prognosis in tumor patients.

### 4.2. Limitations

There are several limitations of this study. We had a small number of research participants, hence, a subgroup assessment is limited.

The used cross-sectional study design has inherited limitations: Different time frames from diagnosis to the present assessment might have influenced our results. Patients with newly diagnosed tumors may have very different resilience levels than patients who have lived for a long time with the disease. Adjuvant treatment and recurrent surgery might further influence it. However, we tried to control for this bias in our multivariate model. Furthermore, our data did not suggest a relevant influence of time since the diagnosis of received CT or RT. Nevertheless, a longitudinal study assessing resilience and its influencing factors should be the next step. Additionally, the current sample might be selective in that patients with a better functional status may more likely have participated in the study.

The main focus of our study was to evaluate the psychosocial situation of patients suffering from a low-grade glioma. Specifically, we aimed to assess resilience as a psychological resource in fighting this disease in this cohort. Biological factors were added only to describe the cohort and were not entered in the respective model calculations. Given the cross-sectional design of our study, certain clinical factors that may influence resilience could not be included in our study (e.g., tumor location, extent of tumor resection or tumor progression and treatment response). Furthermore, all of the diagnoses of the patients were assessed according to the WHO 2016 classification without an assessment of the CDKN2A/B codeletion. Hence, a very small percentage of patients potentially would be upgraded to the WHO grade IV by the 2021 classification.

Even though the ISMI (original version of ISBI adapted for brain tumors) has been used in the past, not just for mental illnesses, but also for rheumatoid arthritis [16], epilepsy [36] and bowel disease [22], the results should be interpreted with caution in our case, given that the measure has not been validated for brain tumors yet. Our results are lacking an exact quantification of the psychological interventions that are performed by specialists in this field.

The influence of occupation on resilience must be interpreted cautiously. Working patients show typically better results in the ECOG, MoCA-Blind tests, and usually their experience of intensive pain was less common. Thus, patients were more quickly able to return to work and had a higher resilience level.

## 5. Conclusions

The patients with a low functional status are at specific risk for having a low level of resilience. Furthermore, the patients in occupation may have an additional burden that could negatively affect their resilience. Both groups should be monitored and provided with support if it is needed. The relevant influence of stigmatization on resilience is a novel finding for patients suffering from a glioma and should be more routinely identified and targeted in the clinical routine.

## Figures and Tables

**Figure 1 cancers-14-05410-f001:**
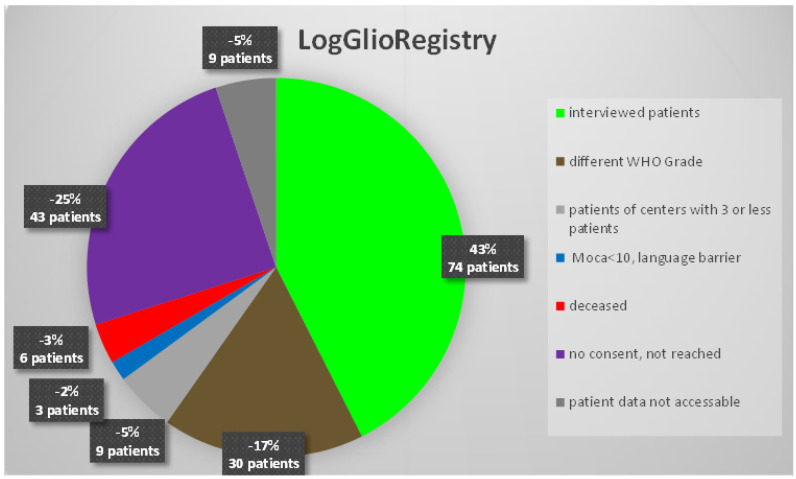
Shows an overview of our study population.

**Table 1 cancers-14-05410-t001:** Demographic characteristics.

				Frequency	Percentage
**sex**	female			32	43.2%
	male			42	56.8%
**educational level**	low			33	44.6%
	high			41	55.4%
**relationship status**	divorced/single			15	20.3%
	married/relationship			59	79.7%
**occupation**	employed			46	62.2%
	sick leave			6	8.1%
	retired			19	25.7%
	unemployed			1	1.4%
	others			2	2.7
**diagnose**	oligodendroglioma	WHO grade 2	IDH wild-type	2	2.7%
			IDH mutant	26	35.1%
		WHO grade 3	IDH wild-type	1	1.4
			IDH mutant	5	6.8
	astrozytoma	WHO grade 2	IDH wild-type	2	2.7
			IDH mutant	25	33.8
		WHO grade 3	IDH wild-type	5	6.8
			IDH mutant	8	10.8

**Table 2 cancers-14-05410-t002:** Adjuvant treatment of patients.

		Frequency	Percentage
**chemotherapy**	no	33	44.6%
	yes	41	55.4%
**radiation**	no	29	39.2%
	yes	45	60.8%

**Table 3 cancers-14-05410-t003:** Results of the questionnaires.

		Frequency	Percentage
≥20		40	54.1%
**MoCa-Blind-Test**	≤19	34	45.9
**pathologic Distress**	no	45	60.8%
	yes	29	39.2%
**RS-13**	low	28	37.8%
	moderate	16	21.6%
	high	30	40.5
**ECOG**	0	42	56.8%
	1	9	12.2%
	2	14	18.9%
	3	8	10.8%
	4	1	1.4%
**ISMI-10**	no to min. stigma	65	87.8%
	slight intern. stigma	6	8.1%
	moderate intern. stigma	3	4.1%
**Pain**	no	55	74.3%
	yes	19	25.7

**Table 4 cancers-14-05410-t004:** Results of the questionnaires in relation to occupation.

	Occupation		No Occupation	
	Mean	N	Mean	N
MoCA-Blind	19.6	46	18.3	28
ECOG	0.3	46	1.8	28
NRS of pain	0.7	46	2.1	28

**Table 5 cancers-14-05410-t005:** Pearson correlations of continuous focal variables with resilience.

	RS13		
	Pearson-Correlations (*r*)	Significance (Two-Tailed)	N = 74
Distress	−0.491 **	<0.001	74
Moca	0.342 **	0.003	74
ECOG	−0.602 **	<0.001	74
Internalized stigma	−0.558 **	<0.001	74
NRS Pain	−0.524 **	<0.001	74
Age	−0.199	0.090	74
Time since diagnosis	0.069	0.561	74
Social support	0.271 *	0.020	74

* *p* < 0.05 (two-tailed); ** *p* < 0.01 (two-tailed).

**Table 6 cancers-14-05410-t006:** Spearman’s ρ correlations of categorial focal variables with resilience.

	RS13		
	Spearman’s ρ	Sig. (2-Seitig)	N
Gender	0.020	0.863	74
Education	0.131	0.265	74
Occupation status	0.244 *	0.039	72
Chemotherapy	−0.112	0.342	74
Radiation Therapy	−0.010	0.934	74
Relationship Status	0.087	0.463	74

* *p* < 0.05 (two-tailed).

**Table 7 cancers-14-05410-t007:** Resilience as criterion and the focal variables as predictors.

Modell	Standardized Coefficient	*T*	*p*	95.0% Confidence Intervall	
	β			LLCI	ULCI
(constant)		4.351	<0.001	45.127	122.022
Moca	0.149	1.307	0.196	−0.440	2.097
ECOG	−0.383	−2.385	0.020	−8.915	−0.779
Internalized stigma	−0.350	−2.742	0.008	−21.444	−3.345
Pain	−0.160	−1.217	0.229	−2.524	0.615
Gender	−0.102	−0.937	0.352	−9.337	3.381
Education	0.0341	0.399	0.691	−4.8289	7.232
Age	−0.112	−0.899	0.372	−0.481	0.183
Occupation status	−0.254	−1.737	0.088	−16.418	1.160
Time since diagnosis	0.012	0.112	0.911	−1.149	1.285
Chemotherapy	0.050	0.270	0.788	−9.247	12.131
Radiation therapy	−0.009	0.050	0.960	−10.774	10.251
Social support	0.026	0.253	0.802	−2.728	3.515
Relationship status	0.037	0.365	0.717	−5.950	8.601

## Data Availability

Data available on request.

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
