# Peer review of "Resilience in Lower Grade Glioma Patients"

_cancers, 2022, doi:10.3390/cancers14215410_

Round 1
Reviewer 1 Report
The present study is interesting because, although the conclusions may be logical, these types of studies are needed to be able to corroborate what seems logical and evaluate it objectively and statistically. However, several concerns need to be addressed:
1. More than 80% of patients in this study had surgical treatment. Have the specimens been analysed by a pathologist to verify the grade of gliomas?
2. When have the patients visit for the follow-up? Have all patients follow up at the same point of the timeline? The authors should specify this information to show that the visits are comparable.
3. Is the visit performed always by the same person? Are the evaluators trained to perform the test?
4. Are the patients evaluated at different time points, or is there only one time to do the comparisons? It would be interesting to have a basal evaluation to compare the follow-up. The initial stage of the patient, either physical or mental, might influence the resilience at the follow-up. Another feature that might be involved in resilience is the state of the patient after surgery. It is not the same that the tumour could be complete resected than only a partial resection. The authors have not mentioned the success of the treatment and it must be discussed. The design of the study could be improved.
5. The percentages in the Table 3 are not the same percentage of the text (sentences 198-201).
6. Sentence 202 is a conclusion that might be in the correlation section. In addition, the percentage of this sentence is not in any table
7. Tables 5 and 6 have no correlation of RS-13 with surgery, marital status or radiation, while in the other tables this information is added. I suggest homogenizing all tables with the same variables.
8. Have the authors the information about the tumour progression or response to the treatment at follow-up? This information is missing in all manuscripts and it would be interesting to take it into account during the study of the correlation between RS13 and the other variables.
9. During the analysis of RS13 correlations, Has the RS13 taken as a numeric and continuous variable or as a categorical variable?
10. There are two tables with the name Table 3. The authors should correct this mistake.
11. There are some p-value in the text that do not correspond to the table, such as sentence 238 and the second table 3
Author Response
Reviewer 1
The present study is interesting because, although the conclusions may be logical, these types of studies are needed to be able to corroborate what seems logical and evaluate it objectively and statistically.
However, several concerns need to be addressed:
- More than 80% of patients in this study had surgical treatment. Have the specimens been analysed by a pathologist to verify the grade of gliomas?
Thank you for this comment. We did not make it clear enough in our methods section. All patients included in our assessment either had biopsy or open surgery to confirm the diagnosis histopathologically according to WHO’s 2016 classification. We have added this information to our methods section as follows:
Histopathlogical diagnosis of patients in this study was used according to WHO 2016 classification. All patients in this study either had biopsy or open surgery that confirmed the diagnosis.
- When have the patients visit for the follow-up? Have all patients follow up at the same point of the timeline? The authors should specify this information to show that the visits are comparable.
All patients have the same routine follow-up according to the LoG Glio protocol which implies a 6-months follow-up for grade II lesions and a 3-months follow-up for grade III lesions. However, in the current study we used a cross sectional study design. This means we telephoned all patients of the included cohort at one time point and thus received one cross-sectional overview of patients who were at different time points in their subjective course of disease. To control for potential confounds in this regard, we used ‘time since diagnosis’ and ‘received adjuvant treatment’ as control variables in our multi-variate model.
Nevertheless, a longitudinal study addressing resilience in low grade glioma patients would be highly important in this context, though it needs many years of prospective study.
We are planning to conduct such a study in our registry in the years to come, but it will take at least a decade to produce valuable data. Therefore, a cross sectional study to exploratively assess influence of resilience is a valuable and economic option to get information about this clinically relevant issue in a first shot.
We included the following paragraph in limitations to emphasize the limitations of a cross sectional study design:
The used cross-sectional study design implies limitations: Different time frames from diagnosis to the present assessment might have influenced our results. Patients with newly diagnosed tumors may differ in their resilience compared to patients having a long course of disease. Adjuvant treatment and recurrent surgery might further influence resilience. We, thus, included these variables to control for a potential bias in our multivariate model. However, our data does not suggest a significant influence of time since diagnosis of received CT or RT on patients’ resilience. Nevertheless a longitudinal study assessing resilience and its influencing factors should be the next step to address this issue.
- Is the visit performed always by the same person? Are the evaluators trained to perform the test?
The phone interview was conducted by E.F. who is an M.D. and has been trained to perform all the tests.
- Are the patients evaluated at different time points, or is there only one time to do the comparisons? It would be interesting to have a basal evaluation to compare the follow-up. The initial stage of the patient, either physical or mental, might influence the resilience at the follow-up.
Patients were evaluated neurologically preoperatively and postoperatively to assesses whether a new neurological deficit occurred. Further, we assessed MOCA (for cognitive deficits) and ECOG (performance status) at time of interview. We discuss this issue in our revised version of this manuscript.
Another feature that might be involved in resilience is the state of the patient after surgery. It is not the same that the tumor could be complete resected than only a partial resection. The authors have not mentioned the success of the treatment and it must be discussed. The design of the study could be improved.
We did not include extent of tumor resection in our study design. We certainly agree with the reviewer that this is a relevant factor for survival. There may also be a potential influence on resilience be it psychologically or clinically. However, it should be noted that this was not focus of our current study. We focused on psychosocial factors influencing patients resilience. Biological factors that certainly have an influence on resilience were not in the focus of our study. We included the following section in limitations:
The main focus of our study was to evaluate the psychosocial situation of patients suffering from a low grade glioma. Especially, we aim to assess resilience as the psychological resources in fighting this disease in this cohort. Biological factors were added only to describe the cohort and were not entered in the respective model calculations. Given the cross-sectional design of our study, certain clinical factors that may influence resilience could not be included in our study (e.g., tumor location, extend of tumor resection or tumor progression and treatment response).
- The percentages in the Table 3 are not the same percentage of the text (sentences 198-201).
We thank the reviewer for this important comment. We have rewritten the results section thoroughly to avoid confusion and hope to have improved this section significantly. We moved all aspects addressing distress to the supplementary documents and added only additional information in the tables. Hence, there should be no redundancies in the text.
- Sentence 202 is a conclusion that might be in the correlation section. In addition, the percentage of this sentence is not in any table
Please see our comment above. We homogenized naming in text tables and methods. Line 202 refers to item 8 from PPQ which we did not display in the tables. We added this information in results and described it more in detail in methods.
- Tables 5 and 6 have no correlation of RS-13 with surgery, marital status or radiation, while in the other tables this information is added. I suggest homogenizing all tables with the same variables.
We thank the reviewer for this important comment and have now homogenized the two correlational tables and the tables displaying results from our multivariate linear regression analyses.
- Have the authors the information about the tumour progression or response to the treatment at follow-up? This information is missing in all manuscripts and it would be interesting to take it into account during the study of the correlation between RS13 and the other variables.
We did not include this information in our study design. At this point our study aims to address psychosocial factors influencing resilience only. Resilience is a very complex construct it merely cannot be addressed in all its aspects. As we pointed out in introduction and discussion. Even genetic factors influence resilience. However, it is an import point for a longitudinal study. We added this to limitations as well. It would be interesting to see whether resilience also influences prognosis.
- During the analysis of RS13 correlations, Has the RS13 taken as a numeric and continuous variable or as a categorical variable?
The RS13 score represents a continuous variable with higher scores indicating a greater extent of resilience. We included RS as a continuous variable in all calculations. However, for a better understanding of the cohort and for comparative reasons we also describe RS 13 in its stepwise characteristics low medium and high.
- There are two tables with the name Table 3. The authors should correct this mistake.
We have adjusted the respective tables accordingly. Thank you for this comment.
- There are some p-value in the text that do not correspond to the table, such as sentence 238 and the second table 3
We thank the Reviewer for pointing this out. Notably, in the text, we refer to Pearson correlations and the respective p-values displayed by Table 5, whereas Table 6 displays Spearnan-Rho correlations and the respective p-values.
We have changed this confusing description and display separate tables for continuous and categorial variables and try to avoid redundancies in the text.
Reviewer 2 Report
In the present manuscript, Fröhlich et al. analyze the correlations of resilience with other psychometric scores such as MoCA, ISBI, PPQ, performance status (ECOG) and other variables such as occupation. The manuscript is well-written and the topic is interesting. However, there are some issues that should be addressed:
Major issues:
1. Baseline characteristics are limited, especially regarding the underlying tumor entities. With the WHO Classifications of 2016 and 2021, grading might have changed and pre-2016 diagnoses may not reflect the biological behavior and clinical prognosis of patients. The authors should provide further data on this issue.
2. Telephone interview may have limitations, especially in estimating performance status as there is also known patient/physician disagreement in estimating ECOG. The authors should discuss this as limitation.
3. Do the authors have any information on how resilience correlates with prognosis?
Minor issues:
1. Please revise language, there are some typos, for example in Figure 1: "diseased" should be "deceased", "no consence" should be "no consent", "patientdata" should be "patient data".
Author Response
In the present manuscript, Fröhlich et al. analyze the correlations of resilience with other psychometric scores such as MoCA, ISBI, PPQ, performance status (ECOG) and other variables such as occupation. The manuscript is well-written and the topic is interesting.
Thank you for your thourough revision and the kind remarks.
However, there are some issues that should be addressed:
Major issues:
- Baseline characteristics are limited, especially regarding the underlying tumor entities. With the WHO Classifications of 2016 and 2021, grading might have changed and pre-2016 diagnoses may not reflect the biological behavior and clinical prognosis of patients. The authors should provide further data on this issue.
Thank you for this important point. A potential upgrading of a small number of patients could occur when assessing CDKN2A/B according to 2021 classification. Yet, we focused on psychosocial factors influencing patients resilience. Biological factors that certainly have an influence on resilience were not in the focus of our study. It is very difficult to include the many different aspects of resilience in a prospective study especially when assessing rare low grade gliomas. We found that addressing the psychosocial situation of these patients in a cross sectional design would be the first step. Further longitudinal studies are needed to address the vast heterogeneity of the biological course of disease.
We added the following information in Methods to better describe the cohort.
Histopathlogical diagnosis of patients in this study was used according to WHO 2016 classification. All patients in this study either had biopsy or open surgery that confirmed the diagnosis.
In limitations we added the following paragraph:
The main focus of our study was to evaluate the psychosocial situation of patients suffering from a low grade glioma. Especially, we aim to assess resilience as the psychological resources in fighting this disease in this cohort. Biological factors were added only to describe the cohort and were not entered in the respective model calculations. Given the cross-sectional design of our study, certain clinical factors that may influence resilience could not be included in our study as tumor location, extend of tumor resection as well as tumor progression and treatment response.
Further, all diagnosis of patients were assessed according to WHO 2016 classification without assessment of CDKN2A/B codeletion. Hence, a very small percentage of patients potentially would be upgraded in WHO grade IV by the 2021 classification.
- Telephone interview may have limitations, especially in estimating performance status as there is also known patient/physician disagreement in estimating ECOG. The authors should discuss this as limitation.
Thank you for this comment. We added this issue to limitations:
We used a phone interview to assess patients performance status. This assessment is limited by the fact that there is no visual on patients performance. Although, in the outpatient setting ECOG performance status assessment is mainly based on anamnestic information often provided by the care givers, plausibility can more easily assessed examining the patient. Hence, there might be a bias in patients assessment. Further, ECOG performance status is not a patient reported outcome measure in general an can be biased by physicians or caregivers perspective.
- Do the authors have any information on how resilience correlates with prognosis?
Thank you for this comment. At time of writing there is no data for a direct influence of resilience on prognosis. There are hints that resilience might influence HRQoL and could by this means influence prognosis. We added this important information for our discussion in the section future perspectives:
It could be shown that resilience has a positive influence on health outcome in breast cancer patients concerning health related quality of life and reduced risk of depression. Further research is need to assess whether there is a direct of indirect influence on prognosis in tumor patients.
Minor issues:
- Please revise language, there are some typos, for example in Figure 1: "diseased" should be "deceased", "no consence" should be "no consent", "patientdata" should be "patient data".
Thank you for this remark. We revised the whole document again and hope to have addressed all typos.
Reviewer 3 Report
The paper addresses an interesting topic and I hope will be extended.
The title indicates adequately the study design.
The abstract provides an informative and balanced summary of the study. The first time it is used MOCA in the abstract have to specify the meaning.
The discussion is fluid and clear.
The words "table 3" and "table 6" are highlighted but is not clear the reason; is it a mistake?
Author Response
The paper addresses an interesting topic and I hope will be extended.
Thank you very much for your positive comments!
The title indicates adequately the study design.
The abstract provides an informative and balanced summary of the study. The first time it is used MOCA in the abstract have to specify the meaning.
We specified the meaning of MOCA in the abstract
The discussion is fluid and clear.
The words "table 3" and "table 6" are highlighted but is not clear the reason; is it a mistake?
We changed the headers of the tables. Thank you for this comment.
Round 2
Reviewer 1 Report
The authors have responded to the comments correctly. The manuscript is having their limitations but now are well explained in the limitations section.
Author Response
Thank you very much for your review and recommendations.
Reviewer 2 Report
The authors appropriately addressed most concerns and improved their manuscript. However, the following points should still be addressed:
1. WHO 2016/2021 diagnoses: The reviewer agrees with the authors that the aim of this study was not to correlate biological factors with resilience. However, histological diagnoses should still be included in table 1 to describe the study cohort even if classified according to the WHO 2016 classification.
2. Still some spelling improvement needed (e.g. extent of resection instead of extent).
Author Response
We have added the diagnosis in the table and improved the spelling in the manuscript as you proposed. Thank you for your recommendations.